# Evaluation of Economic Linkage between Urban Built-Up Areas in a Mid-Sized City of Uyo (Nigeria)

**Etido Essien** [1,*] and **Cyrus Samimi** [1,2,3]

1   Climatology Research Group, University of Bayreuth, 95447 Bayreuth, Germany;
    Cyrus.Samimi@uni-bayreuth.de
2   Institute of African Studies, University of Bayreuth, 95447 Bayreuth, Germany
3   Bayreuth Centre of Ecology and Environmental Research, University of Bayreuth, 95447 Bayreuth, Germany
*   Correspondence: Etido.Essien@uni-bayreuth.de

**Abstract:** Urban growth has transformed many mid-sized cities into metropolitan areas. One of the effects of this growth is a change in urban growth patterns, which are directly linked with household income. Hence, this paper aims to assess the effect of different economic variables that trigger urban built-up patterns, using economic indicators such as city administrative taxes, a socio-economic survey of living standards, household income and satellite data. The regression model was used and adapted, and a case study is presented for the mid-sized city of Uyo in southeastern Nigeria. The result shows sparse built-up growth patterns with numerous adverse effects. Although, there is awareness of the impact of unregulated sparse built-up growth patterns in the literature, little attention has been given to this growth pattern in Africa. The results also show that increases in federal allocation (27%), investment tax (22%), direct tax (52%) and indirect tax (26%) have led to urban expansion into vegetative land and have a causal correlation with different built-up areas. Hence, medium and high-income earners migrate to suburban areas for bigger living space and a lack of basic social amenities affects the land value in suburban areas. They also assist in the provision of social amenities in the neighborhood.

**Keywords:** remote sensing; land-use change; urban growth; socio-economic variables; federal allocation





## 1. Introduction

Urban growth has transformed the built-up patterns of many mid-sized cities into dense urban areas [1]. Urban growth is seen as the physical expansion or uneven growth of undeveloped areas [2]. One of the many effects of this growth is a change in urban growth patterns, which are directly linked with household income [1,3]. These changes differ through time and space [4,5]. At a global scale, the increase in urban populations is triggered by natural population growth and migration towards urban areas for a better standard of living, education, and income [6–9]. These migration patterns of urban growth have complex economic, social, and governance effects [2]. Individuals migrate to urban areas for different reasons [2], including economic, social, and administrative/political reasons [1].

From an economic point of view, factors such as increases in income, the development of road infrastructure and increases in commuting [3,10–13], and from an administrative viewpoint, factors such as lack of proper planning of suburban areas [2] and unregulated land prices have encouraged residents to move to urban areas for better living conditions [1,14]. All of these factors lead to the desire to move and the construction of different residential built-up areas in urban areas that allow for various lifestyles while maintaining the same access and advantages of living in the city center [1]; thus, encouraging further expansion of urban land. This has also enabled rural dwellers to move to cities, to occupy urban areas and create different types of built-up patterns in these areas [1], and to have the same access to social amenities as well as urban life.

Furthermore, a number of studies report that foreign investment, unemployment, economic and population growth are significant factors for shifts in urban development [9,15,16]. Most urbanization studies focus on large cities due to their massive congestion, high population growth, and socio-economic disparities [17–19]. However, the infrastructural needs that usually trigger urban growth in mid-sized cities are generally neglected [14]. The limited research may be attributed to researchers' lack of attention on urbanization in mid-sized cities compared to large cities [17,18]. Hence, understanding urban development in mid-sized cities is crucial because the early pattern of development in mid-sized cities usually affects the patterns of urban growth in big cities [19]. Besides, small and mid-sized cities were found to lead in major urban transformation into metropolitan cities [19]. In this context, Uyo is a prime example of a mid-sized city with a resource-driven economy. The city has experienced urban growth due to infrastructural development, population, and economic growth [14].

Nigeria is one of Africa's largest developing countries with enormous population growth [20]. It is presently experiencing one of the most dynamic urban transformations in the world [20]. After independence in 1960, the country added more than 62.5 million dwellers to its urban areas, with a forecast projecting a further 226 million inhabitants in urban areas by 2050 [20]. A recent government survey showed a rise in rural–urban migration since independence, and unemployment in the rural areas is the primary cause of this migration [21,22]. In Nigeria, robust economic growth in urban areas has functioned as a pull factor over recent years [20]. Even though the economy has experienced a recession, people still move to urban areas [23]. Although, urban and economic growth are frequently entangled, when properly managed, they can bring new developmental strategies for sustainable urban development [18,23]. Understanding the links between socio-economic and urban development helps to provide empirical information as the basis for proper management.

Urban growth has been measured using different tax variables, for example, Wu [24] evaluated the effect of land taxes on urban residential areas, Brueckner and Kim [25] and Peng and Wang [26] examined the effect of land and property taxes on urban residential sprawl, and Ambarwati et al [27] and Tscharaktschiew and Hirte [28] explored transport policies and the effect of improving the public transport network on a city's residential development. In summary, those studies highlight: (i) land taxes positively impact pressure on urban sprawl patterns, (ii) property taxes are efficient tools for urban growth, (iii) land taxes encourage the use of public transport and limit scattered development, and (v) household settlement depends on social amenities and economic income. However, these studies are mostly based on Asian and European cities. As they mostly assume household incomes based on social amenities or property types, this is quite different to Africa due to the lack of social amenities in many households [3,29]. For this reason, our overall objective was to use Uyo as an illustrative example and analyze the city's administrative taxes, a socio-economic survey of living standards, household income, and satellite data to assess the impact of urban growth patterns in the city. We also aimed to answer the following research questions. (i) How do administrative taxes influence the pattern of urban growth? (ii) How do household income and social amenities affect urban land value? (iii) What is the linkage between these socio-economic variables and satellite data?

## 2. Study Area and Materials

### 2.1. Study Area

Uyo is situated in the southeastern part of Nigeria between longitudes $37°50'$ E to $37°51'$ E and latitudes $55°40'$ N to $54°59'$ N (Figure 1). It is the capital city of Akwa-Ibom State and is located in the center of the state. It has a total area of 362 km$^2$ and a low-lying plain with no hills. Uyo is one of the largest commercial cities in southwestern Nigeria after Port Harcourt and Calabar, with an estimated population of 305,961 in 2006 and 429,900 in 2016 [30]. Since the colonial era, the city has been the head of administration and became the state capital in 1987 [30]. This has attracted infrastructural development and prompted

the governmental authority to design a master plan for the city to cope with unplanned urban regeneration [14], However at present, urban development is inconsistent with the city's master plan [14].

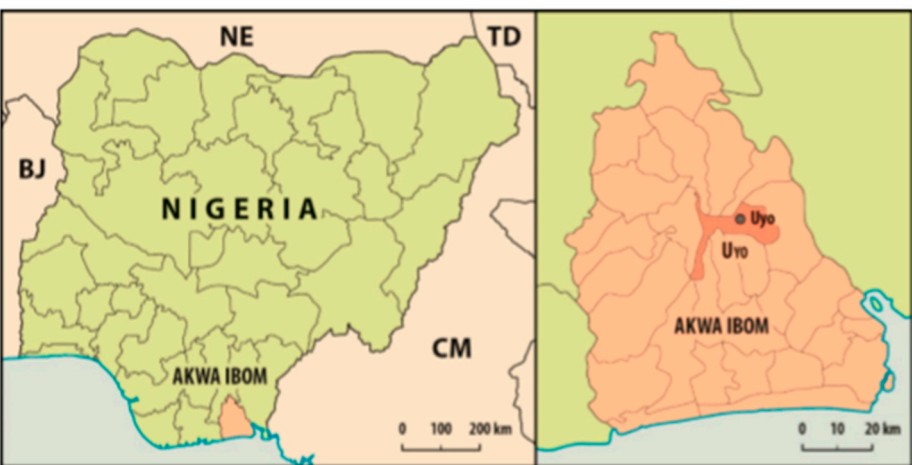

**Figure 1.** Map of Nigeria showing Akwa Ibom State, and map of Akwa Ibom State showing Uyo. Source: Essien & Samimi, 2019.

*2.2. Data Source*

In this study, we used three different approaches to create methodological frameworks for this research. These were based on the socioeconomic variables in the study area and adapted from previous studies [31,32]. Firstly, we drew our data primarily from the socioeconomic survey of living standards and household income from the National Bureau of Statistics of Nigeria (NBS). The socio-economic survey is a national household survey that NBS conducts every five years [33]. We used their recently updated data for our analysis. These data capture all the socio-economic activities in the urban areas in all of the states in the country. We assessed data on household income such as agriculture, production, manufacturing, and the formal sector based on their minimum wage (salary scales). We used these data to calculate and classify household income based on low, medium, and high income [33].

Secondly, we used the city's administrative data on federal allocation, direct and indirect tax, and investment tax (Table 1) from the Ministry of Internal Affairs (MIA). MIA is a government ministry that is responsible for all the internally generated revenues of the state. Our focus was on the administrative taxes of our study area (Uyo) because they continue to increase compared to others socioeconomic variables. These taxes are daily or monthly payments made by small-scale or big enterprises, such as new business tax, property tax, construction tax, and urban development tax. These taxes are charged based on location, the purpose of investment, number of employees and total capital.

Thirdly, we used Rapid Eye images to monitor urban land cover change from 2010 to 2018 in Uyo. Rapid Eye ortho tiles have a 5 m resolution and five identical satellites positioned in a single orbit [34]. The satellite has five multispectral bands (blue, red, green, red edge, and near-infrared (NIR)) [34]. These were geometrically and radiometrically corrected. Thus, sensor-related effects were corrected using sensor telemetry and a sensor model. Spacecraft-related and co-registered effects were corrected using high telemetry and useful ephemeris data [35]. We used Rapid Eye orthoimages with a cloud cover of less than 10% and downloaded eleven images for our study area. We captured the images during the rainy season (June and July) and the core period of vegetation growth. Four images cover the entire study area, whereas seven images cover it to some extent. Post-classification results are often difficult to validate due to no field observation at that time [36]. To do so, we collected ground reference data for training and validation through our fieldwork in the study area. We created the sample so that each class represents the

actual class proportion in the field [37]. We selected a random sample of 320 pixels and a GPS coordinate point at different land cover classes as reference data with detailed vegetation types to ensure that large sample sizes were available for each class and the data were distributed in proportion to their quantity. We overlaid points at random on the 2010 and 2018 multispectral images [38] to identify each pixel's urban land cover type, which was visibly mixed with other urban land cover types, and examined the variation between the two images. We divided the reference data into two parts, 70% for training and 30% for validation data [39] to statistically compare the different urban land cover types and perform an assessment of the accuracy of the classification. We computed the accuracy assessments using confusion matrices based on 30% of our reference data. We used the user's accuracy (UA), producer's accuracy (PA), the Kappa statistic (κ), and urban land cover class percentage as validation metrics for the different land-use classes [39].

**Table 1.** Description of the variables.

| Independent Variables | Dependable Variables | Description |
|---|---|---|
| | Land-use change | Statistical changes that occur in land use over time. |
| Direct tax | | Tax levied directly from individual income or corporate organization by the government. |
| Indirect tax | | Tax levied on the sale of goods by either a manufacturing company or small business. |
| Investment tax | | Tax levied by the government on investors when intending to open a company. This tax depends on the total capital the investor plans to invest. |
| Federal revenue | | An amount paid by the federal government to all the local governments monthly for utilities and projects' maintenance. |

## 3. Methods

### 3.1. Statistical Analyses

The linear statistics model for this research has been described in detail by [31,32,40,41] with regard to its application, standardization, and validation. The model combines statistical data with numerical analysis, is suitable for predicting urban built-up changes, and can also be used to explore different statistical approaches [31,40]. We choose a total of 5 variables to represent the socio-economic growth of the study area, and they have a minimal collinearity of one based on our variance inflation factor (VIF) test. We measured the socio-economic spatial variation in different built-up areas following the example of [31,32,40] as a guide to explore different socioeconomic variables. The selection of this model depends mainly on the likelihood distribution used to model the dependent and independent variables. Besides, the economic data were arranged sequentially (increase in the federal allocation per year). The model has a Pearson probability distribution and a logIn [31]. Pearson residuals revealed no spatial autocorrelation for data in the study area. We assessed the deviance residuals to examine the potential outliers [31]. Variance inflation factors between the independent variables used in these analyses never exceeded two, and most were significantly less, which showed that collinearity was not a significant issue [31]. The model was used to examine the relationship between different urban land cover types (low-density built-up area, medium-density built-up area, high-density built-up area, and government built-up area) and socio-economic variables.

The linear model equation is as follows:

$$\log(Y) = \alpha + \beta \log(X) + e \tag{1}$$

where Y is the urban land cover type (km$^2$), X is the socio-economic variable (naira in local currency), $\alpha$ and $\beta$ are the regression coefficients, and e is the residual error [42]. Regressions were separately computed for each of the urban land cover types, using a coefficient of determination R$^2$ to validate the model's performance and a high value of R$^2$ shows a good model performance [42–44] (Table 1). The statistical significance of regression coefficients-based estimation of t-statistic testing for significant coefficients at *p* > 0.004 was considered as an additional measure for the model's selection [42].

### 3.2. Object-Based Image Analysis

Object-based image analysis (OBIA) is a technique that separates satellite data into significant objects [45]. One of the numerous advantages of using OBIA in image classification is its ability to analyze an object in space rather than a pixel in space [46,47]. One of the common techniques used to generate the object is image segmentation [46]. Segmentation is a method of dividing a satellite image into homogenous objects by merging pixels with similar spectral signatures [48,49]. Segmentation groups pixels with similar features and ensures good image classification results with better accuracy [50]. However, the segmentation algorithm parameters need to be adjusted to get the shape, size, and scale of the resulting object [51], and no generally recognized method is widely used to determine the scale for different environmental applications of OBIA in remotely sensed images [46].

The shape parameter usually determines the spectral homogeneity, while the size parameter balances the object's smoothness [51]. We chose the scale parameter based on how big we wanted our image because the decision on the scale level usually depends on the object's size [51]. We used a weight of 0.2 for the shape parameter to reduce weight on the shape and to produce more homogeneous image segmentation. As well, we adjusted the size parameter to 0.6 to blend the smoothness of the object. After evaluating different scale levels and testing them with different values, a scale level of 20 to 80 was found to be suitable for the study area. We checked the segmented object based on our selected training data, fieldwork knowledge, and differences among the same object classes.

We used K-nearest neighbor (KNN) classifiers to classify the images into different classes and selected samples of training data that represented different classes to reassign each class to the segmented objects [46,51]. This can be achieved in two ways; by giving the classifier the sample of the object training data, and the classifier classifies it based on the nearest neighbors of the sample [46,52]. The advantage of KNN is that it can spectrally separate similar pixels of the same features and assigned segments into different classes with the highest-class confidence value [50]. We used the KNN classifier to identify low-density built-up areas, medium-density built-up areas, high-density built-up areas, government built-up areas, and vegetation (Table 2). We selected these classes based on economic activities in the areas.

**Table 2.** Land use classification type.

| Land-Use Type | Description |
| --- | --- |
| Low-density built-up area | Occupied by either high-income or low-income earners depending on the majority of inhabitants in a neighborhood with a similar type of income. Characterized by high rental fees for businesses and residents, high level of security, a lot of undeveloped land, and near the urban designated area that has most of the social amenities. |
| Medium-density built-up area | Residential area mostly occupied by medium-income earners, near the suburban area and the main road. Affordable rental fees, not too clustered, not so many unsealed streets. |

**Table 2.** *Cont.*

| Land-Use Type | Description |
|---|---|
| High-density built-up area | These are residential areas mostly occupied by low-income earners. Characterized by many informal businesses, clustered houses, cheaper rents, slums, security problems, many unsealed and filthy streets, unstable power supply, and highly polluted. |
| Government built-up area | Characterized by government buildings, offices, new infrastructures, and very few residential buildings owned mostly by old occupants of the area. |
| Vegetation | Low and high vegetation canopy, cropland, football fields, gardens. |

## 4. Results

### 4.1. Economic Growth in Uyo

Economic growth leads back to an increase in the state government revenue. This revenue includes investment tax, direct and indirect tax, and federal allocation (FA). Hence, the historical revenue data extracted from the State Ministry shows that the federal allocation increased annually from 273.665 million Naira (NGN) (USD 98,519) in 2010 to 4.5 billion NGN (USD 1.6 million) in 2018, with an annual growth rate of 27.01% in Uyo. These increases were due to the push for decentralization and for the state to control its natural resource revenue (oil-rent) [53]. Conversely, investment tax increased from 1.4 billion NGN (USD 504,000) in 2010 to 4.1 billion NGN (USD 1.4 million) in 2018, with an annual growth rate of 22.77%. This tax increase was influenced by informal economic growth. Similarly, the direct tax increased from 1.3 billion NGN (USD 468,000) in 2010 to 9.7 billion NGN (USD 3.4 million) in 2018, with an annual growth rate of 53.87%. The increase in the formal sector, such as education has likely offset these tax dynamics. Furthermore, the indirect tax increased from 7.4 million NGN (USD 2664) in 2010 to 4.8 billion NGN (USD 1.7 million) in 2018, with an annual growth rate of 26.65%. Industrialization, such as open markets, increased petty trading, increases in the labor force and an increase in income have acted as the main drivers of this urban growth (Figure 2).

Annual computation of the results shows that increases in federal allocation (27%), investment tax (22%), direct tax (52%) and indirect tax (26%) have led to urban expansion into vegetative land, and have a causal correlation with different built-up areas (Figures 3 and 4, Table 3). According to [33], the investment tax in Nigeria can be increased or decreased based on the investment value. However, from 2010 to 2018, the minimum investment tax in the country, i.e., small-scale enterprises in the urban area, increased by 20% [33]. Additionally, each of the urban communities are allowed to increase the tax to 40%, depending on the building plans of the investor [33]. Small household investments take place among low-income and middle-income earners. These investments tend to grow and expand to big enterprises, significantly encouraging relocation or expansion to different urban built-up areas for larger investment space and customers (Figure 2). This agrees with related studies [1,31,42] that suggest that changes in the different built-up areas, i.e., a 4% increase in the low-density built-up area, an 11% increase in medium-density built-up areas, and a 0.6% increase in high-density built-up areas (Table 4) are triggered by economic growth.

Additionally, real estate (rental) values have increased for middle and high-income households (up to +2%) and decreased for low-income households (up to −0.4%) (NBS, 2020), making the household distribution in the urban areas relatively uneven [14].

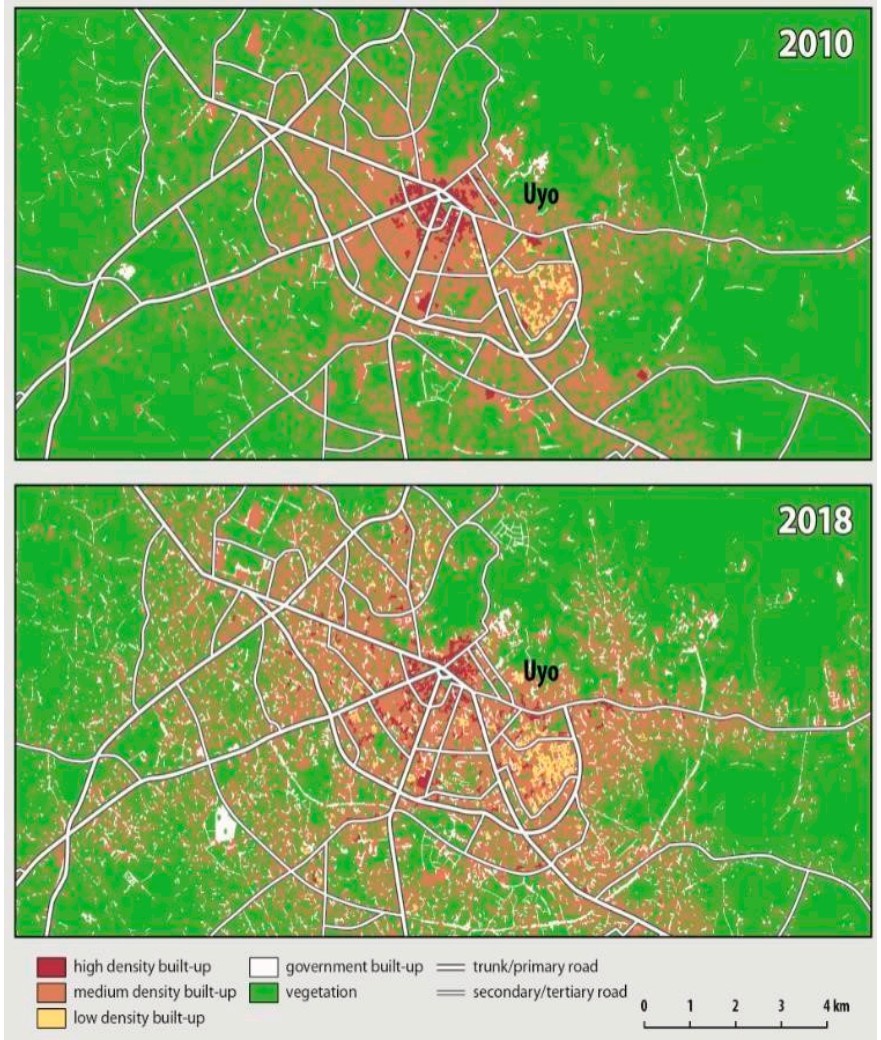

**Figure 2.** Urban land-use land cover changes in Uyo.

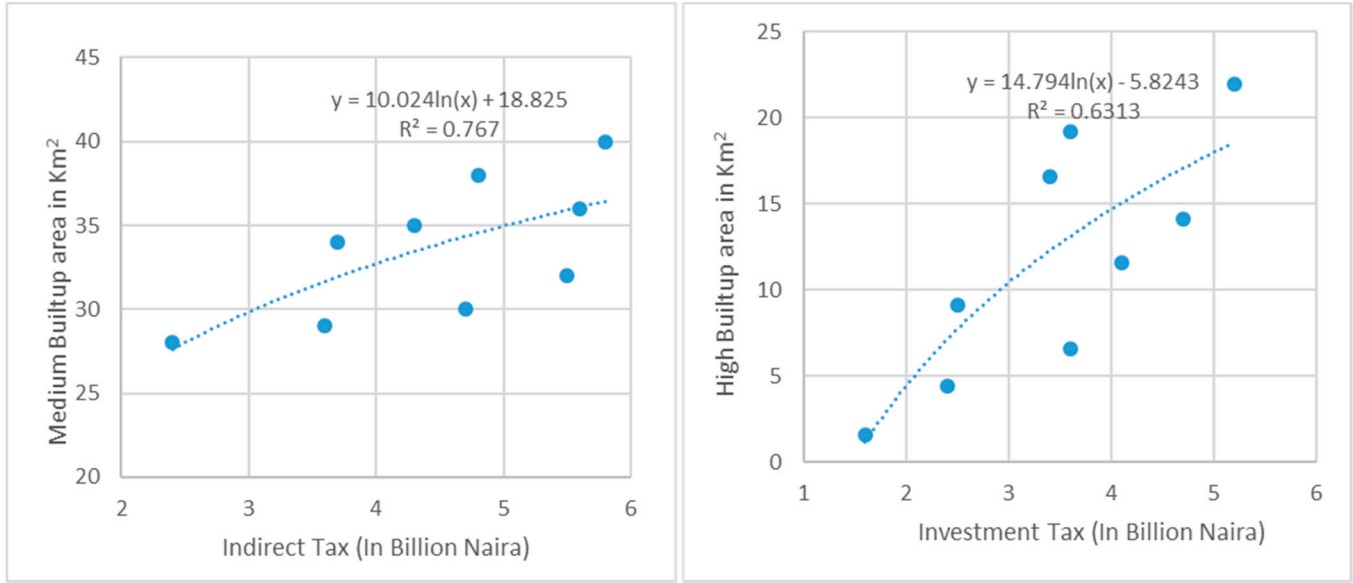

**Figure 3.** Relationship between socioeconomic variables and medium-density built-up areas in Uyo from 2010 to 2018.

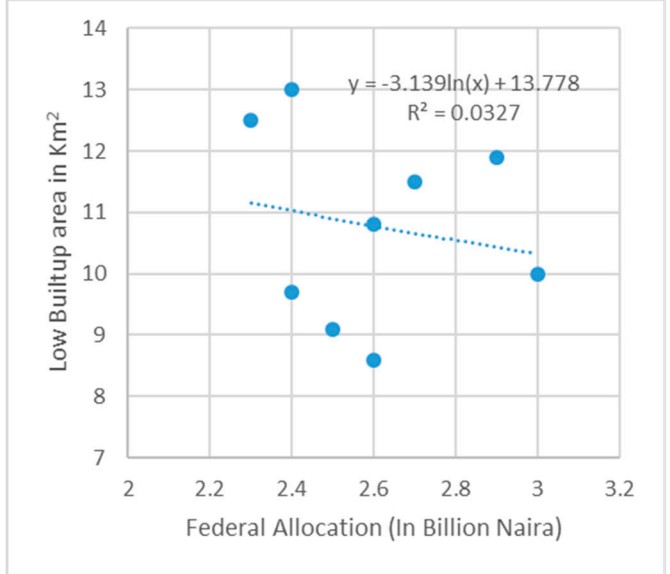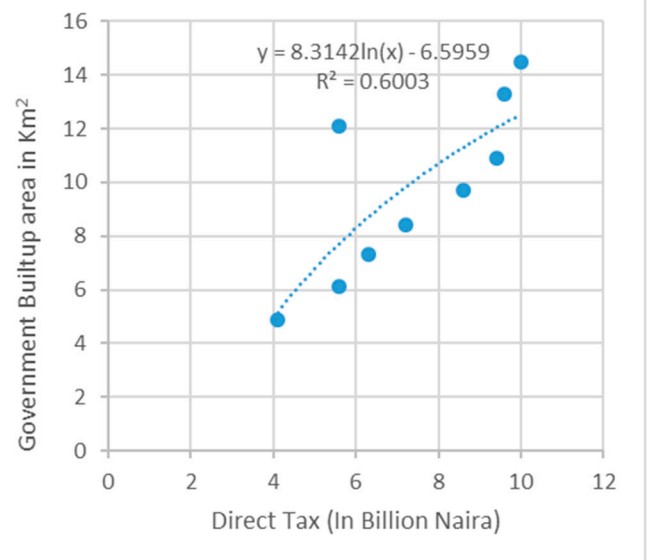

**Figure 4.** Relationship between socio-economic variables and government built-up areas in Uyo from 2010 to 2018.

**Table 3.** Correlation coefficients between socio-economic variables and built-up areas.

| Direct Tax | Indirect Tax | Investment Tax | Federal Allocation |
|:---:|:---:|:---:|:---:|
| 0.03 ** | −0.56 ** | 0.71 ** | 0.62 ** |

** $p > 0.004$, ** $m > 1$.

**Table 4.** Land-use change statistics in Uyo from 2010 to 2018.

| | Area (km²) 2010 | Area (km²) 2018 | Land-Use % | Land-use Change (%) 2010–2018 | Annual Change (km²/yr.) |
|:---|:---:|:---:|:---:|:---:|:---:|
| Low-density built-up | 8.6 | 13.0 | 6.0 | 4.4 | 0.5 |
| Medium-density built-up | 28.4 | 40.3 | 19 | 11.9 | 1.5 |
| High-density built-up | 1.6 | 2.2 | 3.0 | 0.6 | 0.7 |
| Government built-up | 4.9 | 14.3 | 8.8 | 9.3 | 1.2 |
| Vegetation | 43.5 | 26.7 | 63.8 | −16.8 | −2.1 |

*4.2. Social Amenities in the Urban Area*

In our analysis, there was evidence of lack of basic amenities for urban dwellers [33], and according to a recent survey by NBS 2020, there has been an increase in informal settlement across the city. In total, 35% of urban respondents in Uyo have been living in an informal settlement for more than ten years [33]. Few were even born there [29,33], suggesting that despite the government's infrastructure development and claims, the inhabitants of those areas are homeless people [33]. These are urban migrants that cannot afford a better standard of living and have decided to wait for an economic shift. Further, these informal settlements are generally connected to infrastructural development by the government, which demolishes the buildings to build modern infrastructure and create access to basic needs such as water, electricity, and security.

However, informal settlements find it challenging to access these services. A few high-income earners sometimes reside in medium and low-density neighborhoods because land is relatively cheap compared to low-density areas. They buy up large areas, clear it and construct magnificent buildings for residential or commercial purposes with high-end facilities and amenities. Low-income residents sometimes have to pay these high-income neighbors as their alternative suppliers of basic amenities such as water and power because it is relatively cheap and reliable compared to the public services [29]. Half of the city

population has no access to purified drinking water; 50% of urban dwellers drink borehole water, 20% rainwater, 10% surface water, and 12% drink improved bottled water [33]. These statistics (Figure 5) show the variation in the provision of quality water, an essential social need, indicating there is almost no water supply from the government. Nevertheless, 70% of urban households still rely on water sources such as rainwater and boreholes that are widely associated with waterborne diseases such as diarrhea and cholera [33].

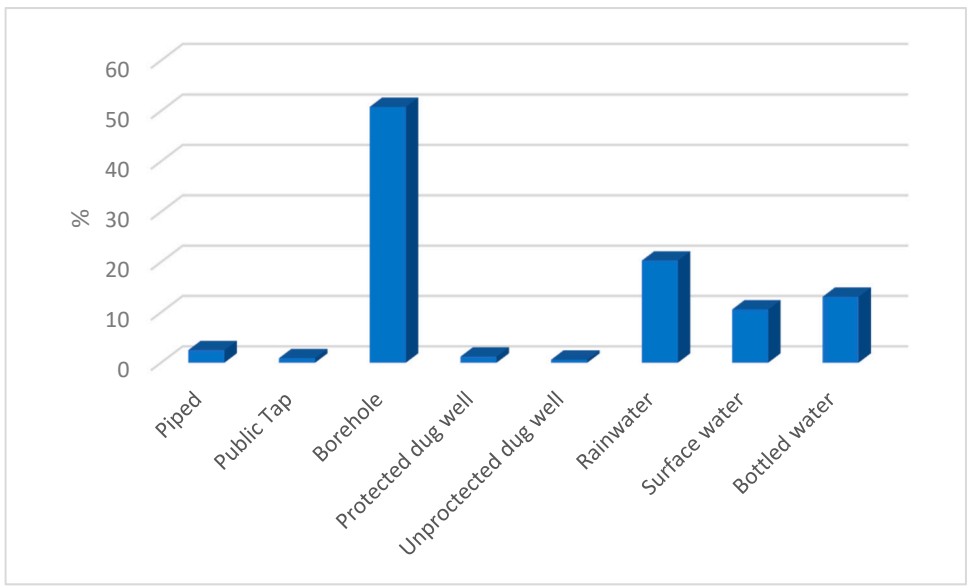

**Figure 5.** Drinkable water in the urban area in Uyo.

Furthermore, no settlement in Uyo has a constant public power supply. The data shows that about 61% of urban dwellers have access to electricity. Still, an estimated billing system, the high cost of electricity, and frequent energy outages have meant that many urban households rely on an improvised alternative power supply, such as gasoline and fuel power-generating plants [29]. These power-generating plants cause constant environmental and noise pollution [29]. Many urban businesses also rely on this source for their daily use. Again, high-income earners switch on their security street lights at night, which serve as an alternative light at night to avert urban crime in the neighborhood [54]. This has encouraged urban communities to sell land at a reduced cost to these high incomes earners with the hope of bringing development and social amenities to their urban neighborhoods. Hence, low and high-income households reside close to each other in the urban area (Figure 2), thus creating a different pattern of urban growth in the city. Although, there are government agencies that regulate urban built-up patterns in the area, their rules are weak and ineffective due to the personal gain they derive from high fines and taxes from urban defaulters [29].

*4.3. Household Income*

According to NBS data at the state level, the monthly household income of low-income workers ranges from N 20,000 to N 40,000 (65%), from N 50,000 to N 100,000 (21%) for medium-income workers, and from N 100,000 to N 500,000 (14%) for high-income workers [33]. The level of income has played a significant role in urban growth in these areas, i.e., a low-income household sometimes sells off their inherited land or property to educate their children and provide for their basic needs [33]. Also, some families are been rendered homeless due to governmental infrastructure development that cuts across their inherited landed property without adequate compensation to property owners [55]. The Nigerian Land Use Act counters the land tenure system of land ownership, and the authority attributes all undeveloped lands to the state [55]. Hence, low-income earners easily sell their land for development purposes to high-income earners to avoid it being taken with

force by the government. This has created visible segregation within urban communities. Low built-up areas are occupied by either high-income or low-income earners (Figure 2), depending on the majority of inhabitants in a neighborhood with a similar type of income. The high-income earners often collectively contribute to infrastructural development [33], hence encouraging the relocation of businesses and migration to new settlements, resulting in an unplanned cluster pattern of urban expansion within these neighborhoods.

Notably, the effect of income inequalities has created different patterns of urban growth among urban dwellers. While 14% live in affluence and have a lot of landed property in different urban built-up areas, 65% live below the average living standard [33]. The authorities have tried to bridge these income inequalities by providing microeconomic loans for low-income earners (non-interest loans to start or support their businesses) and creating jobs for different classes of its urban dwellers [29]. However, most of these jobs are formal and mainly in the educational sector (Figure 6) due to the insignificant number of manufacturing industries in Uyo (Figure 6) [29,33]. The occupants depend mainly on imported goods, and the government relies on the revenue from natural resources for the city's maintenance and development [29]. The authority should support investors in creating more manufacturing and innovative industries to provide more opportunities to low-income earners.

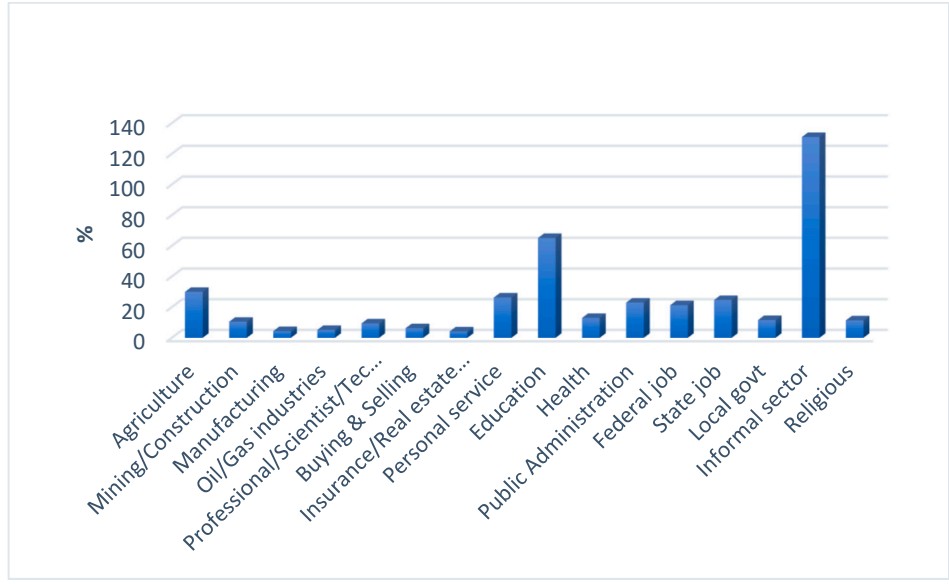

**Figure 6.** Employment sectors in Uyo.

*4.4. Urban Land Cover Change in Uyo*

Land-cover change in Uyo had an overall accuracy of 89%. In 2010, the user and producer accuracy were 89% and 86%, and in 2018, they were 84% and 83%, respectively. Accuracies at the urban land-cover class level were more varied, i.e., the user accuracy for 2010 changed from 88% to 96% for low-density and government built-up areas, respectively. For other classes, a similar pattern was observed, with the high-density built-up area having the lowest user accuracy of 86% in 2010 (Table 5). Equally, vegetation and high-density built-up areas had the highest accuracies of 97% and 99% in 2018, respectively (Table 6). In general, about 5.6% of the land-cover class in Uyo experienced a tremendous transformation within these eight years (2010-2018). Low-density built-up and government areas experienced an increase in area, with increases of 13.0 km$^2$ (4%) and 14.3 km$^2$ (9%), respectively, from 2010 to 2018 (Figure 2). From the interval-based analysis, variation occurred at diverse rates depending on the land-cover type. Medium-density built-up areas also experienced the highest increase of 40.1 km$^2$ (11%) in 2018 compared to its size in 2010. Similarly, the high-density built-up area increased in size over the period by 2.2 km$^2$ (0.6%), with most of these areas converted from vegetative regions. For this

reason, vegetative areas had the highest area loss of 26.7 km$^2$ (16%) between 2010 and 2018 (Table 4).

**Table 5.** Rapid Eye 2010 image classification confusion matrix class.

| Class | Low-Density Area | Medium-Density Area | High-Density Area | Government Area | Vegetation | UA |
|---|---|---|---|---|---|---|
| Low-density Built-up | 34,403 | 0 | 5 | 7 | 0 | 88% |
| Medium-density Built-up | 2 | 1,137,207 | 0 | 9 | 0 | 88% |
| High-density Built-up | 0 | 0 | 64,968 | 0 | 0 | 99% |
| Government Built-up | 0 | 14 | 0 | 199,704 | 0 | 86% |
| Vegetation | 0 | 2 | 0 | 0 | 5,035,248 | 97% |

UA = User Accuracy, kappa statistic = 0.715.

**Table 6.** Rapid Eye 2018 image classification confusion matrix class.

| Class | Low Density Area | Medium Density Area | High Density Area | Government Area | Vegetation | UA |
|---|---|---|---|---|---|---|
| Low Density built-up | 52,261 | 0 | 0 | 9 | 0 | 90% |
| Medium Density built-up | 13 | 1,614,859 | 6 | 0 | 0 | 89% |
| High Density Built-up | 0 | 19 | 89,028 | 4 | 0 | 86% |
| Government Built-up | 0 | 0 | 7 | 573,926 | 0 | 91% |
| Vegetation | 0 | 0 | 0 | 3 | 4,141,456 | 98% |

UA = User Accuracy, kappa statistic = 0.89.

Urban land change has occurred arbitrarily across Uyo for different land-cover types (Figure 2). Vegetation areas have continuously decreased in size as the built-up area has increased in size. Increases in the informal sector have continuously decreased the size of vegetative land areas (Figure 2). For example, the low-density built-up area gradually changed to medium-density built-up and government areas due to increased population density and infrastructural development, such as access road networks and electricity installation [14]. Similarly, medium and high-density built-up areas experienced significant growth across the study period due to urban regeneration [14]. These results show that urban expansion has created different patterns of land cover types, negatively affecting agricultural land (Figure 2).

Tables 5 and 6 provide object-based classification accuracies results. The overall accuracy of the classification was 84.6% in 2010 and 89.6% in 2018. The classification showed a kappa coefficient of 0.715 and 0.824, respectively. The overall accuracy in government and medium-dense areas was 88% and 86%, respectively. However, it was difficult to distinguish the government area from the road because both pixels were classified as one using the K-nearest neighbor OBIA method.

## 5. Discussion

The results of our study demonstrate the use of city administrative taxes, a socio-economic survey of living standards, household income, and satellite data to assess the primary drivers of urban growth in different built-up areas in Uyo (Figure 2). We performed an analysis on the socio-economic variables using an approach adapted from [31,32,40,41]

in the study area. According to the NBS 2020 statistical data for Uyo, low-density built-up areas are situated near the designated urban area. Households in the medium-density built-up areas are near suburbs and the main road, and households in low-density built-up areas live near the urban periphery. These results affirmed that the availability of social amenities is an indicator of economic growth [56], which triggers a continuous increase in all built-up patterns due to urban infrastructural development that is mostly intertwined with economic growth [57].

Results for the link between socio-economic variables and urban built-up areas show a positive correlation with major variations in the urban built-up areas (Figures 3 and 4). This agrees with [58], who suggested that most cities use their revenue for development purposes, especially where most urban dwellers are not farmers. However, our findings provide new insight into some of the reviewed literature [56] that sees land revenue as an efficient tool for steering urban growth and increasing construction in existing built-up areas. Hence, the future effect of this increasing revenue in cities tends to be more predictable and does not depend on any urbanization issue such as housing density [59]. To this end, our study calls for guiding frameworks to be developed when investing city's revenue in infrastructure and social amenities. Revenue can be used to plan and shift urban development to all communities in the city.

Furthermore, our satellite data are not counter to other similar studies on the effects of socio-economic growth on urban growth patterns [31,32,60,61]. However, our land cover classification provides comprehensive urban land cover change data for Uyo in southeastern Nigeria. Previously, urban land cover data for Uyo were only at a 30 m spatial resolution [14,62,63], while studies using high-resolution data (5 m) have been limited in Africa, even at the national level. Constraining urban drivers' understanding of change in various urban land cover classes has resulted in an undefined path for projections on urban land cover change. The 89% accuracy of our map of urban land cover change linked with socio-economic variables makes it appropriate for use as a high multi-temporal resolution map that shows the key drivers of urban land cover change. Furthermore, our research has shown the vital role that informal economic growth in southern Africa has played on urban expansion.

*Data Uncertainty*

The setting of segmentation parameters that control the size, shape, and scale of the object probably affects the classification of the images [51]. However, our different trials regarding this parameter setting affect our classification results because our study area is characterized by different anthropogenic effects. Delineating the boundaries of objects was not easy due to the absence of sharp or hard boundaries.

Land cover types such as high-density built-up areas have no shape or clear boundaries, while government built-up areas were difficult to separate from low built-up areas due to low contrast. However, our accurate delineation of land cover objects might have been time-consuming, but we were able to get homogenous objects for each of the classes. We defined urban land cover classes based on our extensive ground reference data of the area. The training dataset was large enough, so the minority classes were not under-predicted to favor the majority classes in the training data [64]. However, some uncertainties still exist, with a few misclassifications, which mainly occurred in the low-density, medium-density, and government built-up areas (Tables 5 and 6). There was higher uncertainty in the 2018 classification of the urban land-cover class level. There was confusion between medium-density and government built-up areas and confusion between high-density and government built-up areas (Table 6). We believe this effect was very minimal.

## 6. Conclusions

This study assesses urban growth patterns using economic indicators such as socio-economic variables, the socio-economic survey of living standards, household income,

and satellite data in different urban built-up areas. Our results are supported by other studies and are the first for sub-Saharan Africa. The results for the city of Uyo shows that (i) an increase in socioeconomic revenue has led to urban expansion and there is a causal correlation with different built-up areas, (ii) a lack of basic social amenities decreases the value of land in the suburban areas, and (iii) medium and high-income earners often migrate to suburban areas for bigger living space. Although, this urban growth pattern is widely recognized as a problem for urban planners, these results contribute to creating awareness of the weak urban planning laws in the study area and in other African cities. They could also guide the monitoring of urban growth in different urban land cover classes. Similarly, our study could also be used to redesign the city and aid in sustainable urban management, and help the authorities globally with spatial planning to monitor unplanned urban growth.

**Author Contributions:** Conceptualization, E.E.; Data curation, E.E.; Formal analysis, E.E.; Methodology, E.E.; Supervision, C.S. All authors have read and agreed to the published version of the manuscript.

**Funding:** This research was funded by the German Academic Exchange Service, grant number 91666076.

**Institutional Review Board Statement:** Not applicable.

**Informed Consent Statement:** Not applicable.

**Acknowledgments:** The authors are grateful to the editor and reviewers for their valuable comments and suggestions.

**Conflicts of Interest:** The authors declare no conflict of interest.

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
