# Peer review of "Evaluation of Economic Linkage between Urban Built-Up Areas in a Mid-Sized City of Uyo (Nigeria)"

_land, doi:10.3390/land10101094_

Round 1
Reviewer 1 Report
This is an interesting manuscript and focuses on a topic of interest for the readership of Land journal. Nevertheless, it has some unclear issues. Please see the following list of comments:
- the objectives of your study should be clearer in the introduction by indicating the general objectives and the specific objectives of your study.
- section 2.1 You should include a map illustrating the case study area.
- Please include a methodological framework.
- Table 1: how were these independent variables identified? why did you choose these variables and not others? You stated that “they have minimal collinearity”, but how much? what methods/techniques did you use?
- why did you use the methodology adopted by (Kai-y & Hao, 2012., 2013)? It needs to be justified and it needs more explanations regarding alternative approaches.
- what are the limitations of your study?
- In the conclusion section you should expand and emphasise the contribution and implication of your results for spatial planning.
Author Response
- the objectives of your study should be clearer in the introduction by indicating the general objectives and the specific objectives of your study.
Response: Thanks for your comment, we have made it to be clearer now. See number 98
- section 2.1 You should include a map illustrating the case study area.
Response: Thanks, we have inserted the map as suggested. See number 126
- Please include a methodological framework.
Response: Thanks for noticing it, we have outlined sequential methodological approached as suggested. See number 132
- Table 1: how were these independent variables identified? why did you choose these variables and not others? You stated that “they have minimal collinearity”, but how much? what methods/techniques did you use?
Response: Thanks for your comment, In the socioeconomic data of the city, the independent variables were chosen based on their continuous increase compared to other socioeconomic variables. They have a minimal collinearity of one based on our variance inflation factor (VIF) techniques test. See number 189.
- why did you use the methodology adopted by (Kai-y & Hao, 2012., 2013)? It needs to be justified and it needs more explanations regarding alternative approaches.
Response: Thanks for your comment, we adopted this method because the model combines statistic data into numerical analysis, and are suitable for predicting urban built-up changes, and can as well be used to explore different statistical approaches. See number 185
- what are the limitations of your study?
Response: Thanks for your comment, most of my limitation are outlined in data uncertainty. See number 476
- In the conclusion section you should expand and emphasise the contribution and implication of your results for spatial planning.
Response: Thanks for your comment, our aim was to make the conclusion to be concise, but I have added more to it as you suggested. See number 505
Reviewer 2 Report
The article deals with an interesting topic and the research questions are valuable. The attempt to include data from satellite images in the analyzes should be assessed very highly. I am not able to verify if the photos are correctly processed (I do not know the topic and hope that other reviewers will verify it), but the results seem to be correct. Unfortunately, I have a lot of comments on the rest of the article.
Introduction:
I admit that I got lost with the authors' arguments. Firstly, the authors write that there is a tendency to migrate to cities because it enables people to live in better conditions. In the second part of the introduction, the authors write that the population is moving to suburban areas - also in order to live in better conditions… .. What processes are the authors writing about? Is it urbanization or suburbanization? My doubts also result from the fact that the term "Urban growth" has not been defined. The authors describe how urban growth is measured, but do not explain how they define it.
Data source
I am asking for a better justification why taxes are good measures of urban development. The increase in the built-up area is also a measure of urban development. Why, then, the authors explain the growth of the built-up area with the amount of taxes?
Result
I have serious doubts whether the research assumptions and results are correct:
"Increase in administrative taxes has led to urban expansion and it has a casual correlation with different built-up areas" Or is it the other way around? In my opinion, it is urban expansion that led to increase in administrative taxes.
The authors' research shows that the city is developing thanks to high taxes. In my opinion, this conclusion contradicts logic and other research.
I have serious doubts whether the variables in the regression models have been well identified. The main question is what is the cause and what is the effect? In my opinion, the dependent variable should be the different taxes, and their amount can be explained by the increase in built-up areas… And the conclusions will be different: urbanization and urban growth lead to an increase in the income of cities.
Please explain why the variable Y is the built-up area and why it is explained by the amount of taxes.
Minor Notes:
Economic growth has transformed many mid-sized cities into metropolitan areas " Do the authors really mean the process of metropolisation? Or maybe is it about urban sprawl ??? Please see the definition of the metropolisation process
Figure 5. Electricity usage in Uyo - The drawing is redundant.
Please correct the description of tables 4a and 4b…. One table’s title is below the table, the other above.
The authors should subject the article to thorough verification and respond to my questions.
Author Response
Introduction:
I admit that I got lost with the authors' arguments. Firstly, the authors write that there is a tendency to migrate to cities because it enables people to live in better conditions. In the second part of the introduction, the authors write that the population is moving to suburban areas - also in order to live in better conditions… .. What processes are the authors writing about? Is it urbanization or suburbanization? My doubts also result from the fact that the term "Urban growth" has not been defined. The authors describe how urban growth is measured, but do not explain how they define it.
Response: Thanks for noticing this. It has been corrected. See number 53,28
Data source
I am asking for a better justification why taxes are good measures of urban development. The increase in the built-up area is also a measure of urban development. Why, then, the authors explain the growth of the built-up area with the amount of taxes?
Response: Thanks for your comment. Increase in socioeconomic activities triggers increase in built-up areas, and taxes and other socioeconomic variables in the study areas shows continues increase in revenue. This increase in revenue attracts urban development as you said. See number 255-382
Result
I have serious doubts whether the research assumptions and results are correct:
"Increase in administrative taxes has led to urban expansion and it has a casual correlation with different built-up areas" Or is it the other way around? In my opinion, it is urban expansion that led to increase in administrative taxes.
Response: Thanks for your comments, we have edited it. Increase in socioeconomic revenue attract urban development. See number 505
The authors' research shows that the city is developing thanks to high taxes. In my opinion, this conclusion contradicts logic and other research.
Response: Thanks for your comment, we mean high tax revenue and not high taxes due to increase in tax. In the introduction I have cited other related studies that show increase in tax revenue affects urban growth in developed countries. See number 87. But we integrate other socioeconomic data with tax data in our study area in context of developing country to assess urban growth pattern.
I have serious doubts whether the variables in the regression models have been well identified. The main question is what is the cause and what is the effect? In my opinion, the dependent variable should be the different taxes, and their amount can be explained by the increase in built-up areas… And the conclusions will be different: urbanization and urban growth lead to an increase in the income of cities.
Response: Thanks for your comment, “urban growth leads to an increase in the income of cities” you are right, but this normally happen in big cities. However, in mid-sized cities, socioeconomic revenue and urban infrastructural development leads to urban growth. This can be seen in our urban classification map. The map shows urban growth that occurred in 2010 and 2018. See number 418
Please explain why the variable Y is the built-up area and why it is explained by the amount of taxes.
Response: Y (built-up) is the dependable variable because increase in socioeconomic revenue(taxes) X increases infrastructural development and affects urban land use change. See number 293
Minor Notes:
Economic growth has transformed many mid-sized cities into metropolitan areas " Do the authors really mean the process of metropolisation? Or maybe is it about urban sprawl ??? Please see the definition of the metropolisation process
Response: Thanks for this comment, we have corrected it. See number 27
Figure 5. Electricity usage in Uyo - The drawing is redundant.
Response: Thanks for your comments, we have removed it. See number 341
Please correct the description of tables 4a and 4b…. One table’s title is below the table, the other above.
Response: Thanks for this comment, we have corrected it. See number 433
Round 2
Reviewer 1 Report
Dear authors,
The manuscript improved with the revisions. However, it still has some flaws:
-You need to identify the research gaps for your research and clarify what are its innovative contributions to science.
- Line 101: iii) what is the linkage of these socio-economic variables with satellite data?
please be clearer on this issue. what do you mean by satellite data?
Line 158: “We selected a random sample of 320 pixels”. why 320? this has to be justified
Line 179: “they have minimal collinearity” but how much? this has to be shown in the results section.
Author Response
Thank you so much for taking your time again to improve this paper, I appreciate it.
The manuscript improved with the revisions. However, it still has some flaws:
-You need to identify the research gaps for your research and clarify what are its innovative contributions to science.
Response: Thanks for your comments, we have shown our research gaps and contribution to science in numbers 58-70, 95- 98, 474-484
- Line 101: iii) what is the linkage of these socio-economic variables with satellite data?
Response: Thanks for your comment, we show the linkage by using an economic model (linear regression) to examine the effect increase in socioeconomic variables have on urban built-up areas. See figure 3 in number 303
please be clearer on this issue. what do you mean by satellite data?
Response: Thanks for your comment. By Satellite data, we mean remote sensing data. We have explained it in detail in numbers 156 and 217
Line 158: “We selected a random sample of 320 pixels”. why 320? this has to be justified
Response: Thanks for your comments, we have edited it see number 171
Line 179: “they have minimal collinearity” but how much? this has to be shown in the results section.
Response: Thanks for your comments, we have edited it and show the correlation coefficients for each of the variables, see table 3 in see number 286
Thank you once again we are grateful for all your comments, I hope I have answered all your questions.
Reviewer 2 Report
Thank you for the answers and the changes that were made. Nevertheless, I am still not convinced that the authors are right about the relation: tax revenues ----> urban development.
I am not familiar with the specifics of developing countries, but I still believe that the model has some false assumptions. On the basis of which economic theory did the authors construct the model? There is at least a dozen of them…. In my opinion, references to other articles are not enough.
The city is developing thanks to new residents, new companies and investments - mainly private ones. It can grow thanks to innovation (which is less the case for developing countries). The increase in tax revenues is only a consequence of urban development, not the cause !!!! The increase in tax revenues can only indirectly contribute to the development of cities - in a situation where city revenues are invested in the development of infrastructure.
The identified dependence can also be explained in a different way: tax revenues are growing because new residents are moving into the city, whose incomes are growing. Rising incomes drive demand, etc.
Author Response
Thank you so much for taking your time again to improve this paper, I appreciate it.
Thank you for the answers and the changes that were made. Nevertheless, I am still not convinced that the authors are right about the relation: tax revenues ----> urban development.-
Response: The city major source of revenue comes from oil revenue (Federal allocation). This revenue is mostly used for urban infrastructural development. See 246
I am not familiar with the specifics of developing countries, but I still believe that the model has some false assumptions. On the basis of which economic theory did the authors construct the model? There is at least a dozen of them…. In my opinion, references to other articles are not enough.
Response: Thanks for your comments, like you said, there are dozens of them, but we tested different models to know the one that fit and correlates with our variables and how well the model explains the linkage of our variables before we choose it. We used linear regression model because it fit our model. We showed the model equation and described each of the model parameters and we have added the coefficient of the correlations between different built-up areas. see in number 173, 286
The city is developing thanks to new residents, new companies and investments - mainly private ones. It can grow thanks to innovation (which is less the case for developing countries). The increase in tax revenues is only a consequence of urban development, not the cause !!!! The increase in tax revenues can only indirectly contribute to the development of cities - in a situation where city revenues are invested in the development of infrastructure.
Response: Thanks for this comment, our study area is a developing mid-sized city, where most of its revenue is reinvested in the city for infrastructure development purposes and the major source of the city revenue comes from oil revenue (federal allocation) see number 246. This development is linked to increased in tax revenue as you said, this revenue increase spread development into different urban built-up areas. However, in our study, we didn’t use only tax data but we integrate different socioeconomic data such as national survey of informal settlement in cities in Nigeria, see number 346, remote sensing data to monitor changes with time, see number 425.
The identified dependence can also be explained in a different way: tax revenues are growing because new residents are moving into the city, whose incomes are growing. Rising incomes drive demand, etc.
Response: Thanks for your suggestion, we have edited it, see number 272
Thank you once again we are grateful for all your comments, I hope I have answered all your questions.